# Breast-Feeding Protects Children from Adverse Effects of Environmental Tobacco Smoke

**DOI:** 10.3390/ijerph16030304

**Published:** 2019-01-23

**Authors:** Hanns Moshammer, Hans-Peter Hutter

**Affiliations:** Department of Environmental Health, Center for Public Health, Medical University of Vienna, 1090 Wien, Austria; hans-peter.hutter@meduniwien.ac.at

**Keywords:** breast-feeding, passive smoking, lung function, early life exposures, protective factors

## Abstract

In a cross-sectional study on 433 schoolchildren (aged 6–9 years) from 9 schools in Austria, we observed associations between housing factors like passive smoking and lung function as well as improved lung function in children who had been breast-fed. The latter findings urged the question of whether the protective effects of breast-feeding act on environmental stressors or if they act independently. Therefore, the effect of passive smoking on lung function was stratified by breast-feeding. The detrimental effects of passive smoking were significant but restricted to the group of 53 children without breast-feeding. Breast-feeding counteracts the effect of environmental stressors on the growing respiratory organs.

## 1. Introduction

We previously reported on 433 schoolchildren (aged 6–9 years) from nine schools in Austria. We found associations between indoor air quality in the schools and lung function of the children [1], between organophosphate flame retardants in the dust of classrooms and cognitive performance of the children [2], and between household factors like passive smoking reported in a questionnaire by the parents and lung function [3]. We also observed better lung function in children who had been breast-fed. The household factor that most strongly reduced lung function was “number of smokers in the household”.

A protective effect of breast-feeding on lung function later in childhood has been reported in several studies, e.g., those in References [4,5]. However, the underlying mechanisms are still not clear. To further the knowledge about the mode of action, we set out to test the hypothesis that breast-feeding acts on and interacts with environmental stressors. As opposed to independent effects, this would indicate a mechanism affected by breast-feeding and environmental stressor jointly. 

To that end, an analysis of the effects of “number of smokers” on lung function was performed stratified by reported breast-feeding. In addition, the effect of breast-feeding was investigated for different durations of breast-feeding. In this analysis, “number of smokers” served as an indicator of childhood environmental tobacco smoke (ETS), which is the strongest environmental predictor of lung development.

## 2. Materials and Methods 

In a nationwide project, nine Austrian primary full-day schools from Vienna, Lower Austria, Styria, and Carinthia were selected and included in a cross-sectional study. Indoor air quality was assessed in the schools and linked to details of construction and also to siting of the school buildings. Chemical analyses were performed in the air (gaseous pollutants), in the airborne dust (PM2.5), and in samples from settled dust. 

In 2–4 classes (1st and 2nd grade), children were invited to participate in health examinations. Parents provided informed consent. Children were instructed about the planned procedures and could abstain from any examination. Headmasters and teachers supported the study and distributed information material, consent forms, and questionnaires to the parents in advance. The study was approved by the Ethics Committee of the Medical University of Vienna and the Vienna General Hospital (permit no. EK 492/2006).

Here, we report on the effects of breast-feeding as reported by the parents (usually the mother) and on passive smoking on lung function. Information on breast-feeding was obtained both in a binary format (“ever breast-feeding: yes/no”) and as an ordinal variable (Table 1). Passive smoking was asked through several questions: “How many cigarettes are usually smoked in the home per day?”/“How many smokers are in the household?”/“Does the mother smoke?”/“Did the mother smoke during pregnancy?”/“Did the mother smoke while breast-feeding?”/“Does the father smoke?”

Lung function testing was performed according to the protocol of the American Thoracic Society [6], except for the minimum exhalation time of 6 s (which is not feasible for children). Both volume and flow parameters were documented: Forced vital capacity (FVC), forced expiratory volume in the first second (FEV1) and in the first half second (FEV0.5), peak expiratory flow (PEF), maximal mean expiratory flow (MMEF), maximal expiratory flow at 25% of FVC (MEF25), maximal expiratory flow at 50% of FVC (MEF50), and maximal expiratory flow at 75% of FVC (MEF75). All lung function tests were conducted between 8:30 and 12:30 a.m. 

Age (in years), height (in cm), weight (in kg), and sex of each participating child was noted. For each lung function parameter, a linear regression model was developed that included the exposure of interest, all anthropometric measures, also including weight squared [7]. Anthropometric measures were removed step by step from the model (starting with that with the highest *p*-value) if they did not contribute to the model fit (*p* > 0.2 or *p* > 0.1 and no relevant effect on the point estimate of the exposure effect). Different measures of ETS exposure (number of cigarettes, number of smokers, maternal smoking at different time points, current paternal smoking) were included as an independent variable, in addition to breast-feeding and an interaction term between both. All statistical analyses were done with STATA 13.1 SE (StataCorp, 4905 Lakeway Drive, College Station, TA, USA, 2013) [8]. 

## 3. Results

Table 1 presents the sample of 433 schoolchildren. Not all questions were answered by all parents. Not all children provided valid FEV1 results. Therefore, FEV0.5 was also investigated.

All ETS parameters reduced lung function values. While 133 mothers (of 431) reported current smoking, only 94 (of 431) reported smoking during the first year of the child, only 42 (of 432) during breast-feeding, and only 52 (of 433) during pregnancy. Although early life ETS exposure (pre- and post-birth) is a strong predictor of lung function [9] and this finding was also confirmed in this sample [3], the numbers were too small to investigate interaction effects meaningfully. For example, only 17 mothers reported smoking in the first year of the child and never breast-feeding, 11 smoking during pregnancy and never breast-feeding. The number of smokers in the household was more evenly distributed and, therefore, was selected as the best marker of ETS exposure for the analysis of protective effects of breast-feeding. 

The detrimental effects of smokers in the household were significant but restricted to the group of 53 children without breast-feeding (Figure 1). After controlling for age, sex, height, and weight, lung function improved with breast-feeding and that effect grew stronger with the duration of breast-feeding (Figure 2).

Breast-feeding counteracts the effect of environmental stressors like environmental tobacco smoke on the growing respiratory organs. Although most of the children were breast-fed and, thus, the power to detect an effect of ETS exposure would have been sufficient, in general, no effect of ETS could be observed in that group. The well-known impact of ETS on poor lung function that was evident in the overall group was driven solely by the fewer children without breast-feeding (Figure 1). The reverse was found not to be true (data not shown): Breast-feeding (and especially longer breast-feeding) improved all lung function parameters no matter whether there was a smoker in the household or not. A clear dose–response function with better lung function with increasing duration of breast-feeding was visible (Figure 2).

## 4. Discussion

Breast-feeding likely protects not only from the adverse effects of ETS, but also from other ambient influences. Other environmental causes of poorer lung function, like, e.g., air pollution, were visible in that population [3] but generally too weak to meaningfully allow the investigation of interaction effects. However, more generally protective effects from breast-feeding are to be expected also regarding these other and many more factors, as maternal milk has antioxidant and anti-inflammatory properties, influences gut microbiome development and supports the maturation of the immune system. This might explain why there is no clear impact on smoking in the household on the beneficial effects of breast-feeding.

Other studies have demonstrated the beneficial effects of breast-feeding on lung function until the age of 10 [4] or even 16 [5]. The latter study found beneficial effects only in children whose mothers were not asthmatics. In addition, Soto-Ramírez et al. [10] noted the beneficial effect of breast-feeding at 10 years of age but no longer at an age of 18. In line with Guilbert et al. [5], Waidyatillake et al [11], studying a birth cohort with a family history of atopy, did not find beneficial effects of breast-feeding at age 12 and 18. These results are contradicted by Guilbert and Wright [12], who also found beneficial effects in children of asthmatic mothers. 

To sum up, most studies demonstrate the beneficial effect of breast-feeding on lung growth, although some of these studies point to weaker effects or even no effects in the case of atopy. To our knowledge, no published study so far has studied the interaction of breast-feeding with environmental exposures. However, we acknowledge that the re-analysis of our own data was encouraged by an unpublished study from China (personal communication by Chuan Zhang). That Chinese study examined 6740 children aged 7 to 14 years and found stronger associations of air pollution with lung function among non-breastfed children than among breastfed children. 

One limitation of our study is the sole reliance on questionnaire data regarding ETS exposure [13]. Validation of this information could have been performed by biomonitoring of cotinine in the hair or in urine, serum or saliva [14] of the children. However, chemical analyses were too costly and sample collection (especially regarding serum) was deemed too complicated and might have lead to a reduced participation rate. Additionally, biomonitoring results have shortcomings, as cotinine levels in children are also affected by the time the child has recently spent around smokers [15]. Past exposure and exposure in early life cannot be assessed through biomonitoring. The clear and expected effects of the ETS parameters on lung function give credence to the reported exposures.

Household smoking and breast-feeding could both be associated with some socioeconomic factors. When both are strongly correlated with each other, the assessment of an interaction or even the joint analysis of both factors would not have been possible. Fortunately, correlation was low. The rank correlation between the duration of breast-feeding and number of smokers was –0.22 and with the number of smokers –0.14 only.

## 5. Conclusions

Interactions are hard to demonstrate in epidemiological studies because of power constraints. Very large data bases are needed when the primary effects are already weak, which is often the case with environmental exposures and life-style factors. This might be one reason the interaction between breast-feeding and environmental exposures has not been investigated in depth in scientific literature. This is a pity, because more studies of this kind would shed light on the possible mechanisms of (observed) effects. 

Breast-feeding improves lung function in children and that effect can be demonstrated up to school age. It seems that some—if not all—of this effect is caused by counteracting the adverse effects of various environmental exposures that might ultimately lead to inflammatory reactions in the lung and to an increased risk of respiratory infections. Anti-inflammatory and anti-infective compounds in breast milk are likely responsible for the observed beneficial effects.

## Figures and Tables

**Figure 1 ijerph-16-00304-f001:**
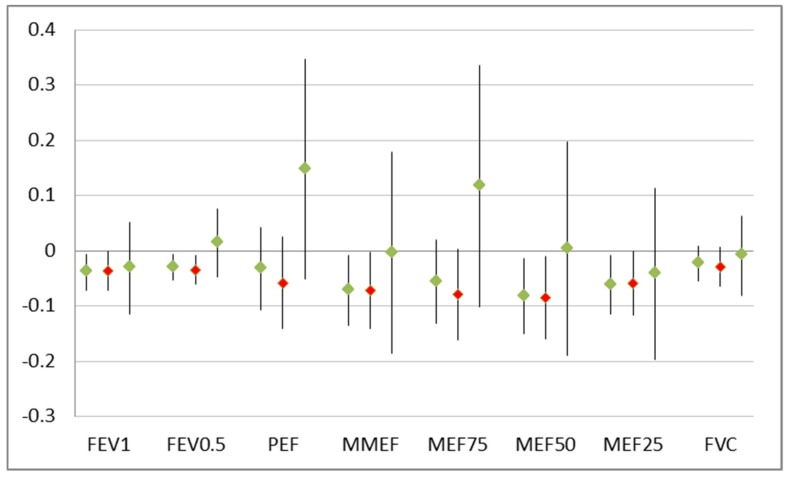
Results of linear regression on number of smokers on lung function parameters: Change in liter (FVC, FEV1, FEF0.5) or in liter/sec (other parameters) per smoker. Point estimates and 95% confidence interval: All 433 children (**left**), 53 children without breast-feeding (red marker, **middle**), and 380 breast-fed children (**right**).

**Figure 2 ijerph-16-00304-f002:**
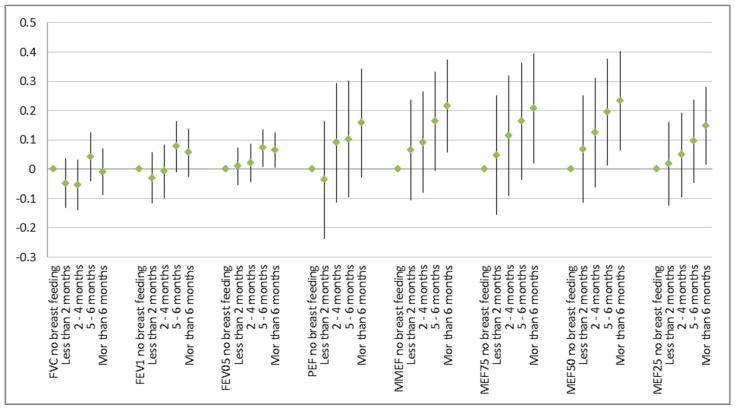
Effect of duration of breast-feeding on lung function in comparison to no breast-feeding: Point estimates and 95% confidence intervals in liter (FVC, FEV1, and FEV0.5) or in L/s.

**Table 1 ijerph-16-00304-t001:** Description of the sample.

Parameter	Number	Mean +/− Std.Dev.
Age (years)	433	7.3 +/− 0.6
Height (cm)	433	127.0 +/− 6.9
Weight (kg)	433	27.5 +/− 6.1
FVC (L)	433	1.6 +/− 0.3
FEV1 (L)	337	1.6 +/− 0.3
FEV0.5 (L)	433	1.2 +/− 0,2
PEF (L/s)	433	3.0 +/− 0.7
MEF75 (L/s)	433	3.0 +/− 0.7
MEF50 (L/s)	433	2.3 +/− 0.6
MEF25 (L/s)	433	1.3 +/− 0.4
MMEF (L/s)	433	2.2 +/− 0.5
Sex	433	
Male	215	
Female	218	
Number of Smokers	420	
0	207	
1	147	
2	64	
3	2	
Duration of breast-feeding	426	
Never	53	
Less than 2 months	79	
2–4 months	76	
5–6 months	88	
More than 6 months	130	

FVC: Forced vital capacity; FEV1, FEV0.5: Forced expiratory volume in the first second and in the first half second, respectively; PEF: Peak expiratory flow; MEF75, MEF50, MEF25: Maximal expiratory flow at 75%, 50%, and 25% of FVC, respectively; MMEF: Maximal mean expiratory flow.

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
