# Peer review of "Breast-Feeding Protects Children from Adverse Effects of Environmental Tobacco Smoke"

_ijerph, 2019, doi:10.3390/ijerph16030304_

Round 1

Reviewer 1 Report

one important problem is the byass due to self declaration by questionnaire

perhaps the authors could discuss about advantadges of biomarkers of environmental tobacco exposure to nicotine

what about other important factors as LTRI?

families smoking at home could have some socioeconomic problems

deleterious effects on lung function were due to prenatal or to postnatal exposure to smoke?

breast milk have an obvious positive general effect, but only breast milk o it is associated with other factors?

Author Response

see attached word document!

Reviewer 2 Report

1, The introduction should provide more background about the research questions and include relevant references. 2, I would encourage the authors to strengthen the Material and methods by more closely mapping the analytical methods to their research questions (instead of just offering many different analyses and findings). 3, I would encourage the authors to more clearly specify their models (outcomes, testable covariates, confounders by name). 4, I would encourage the authors to show the P-value or significant level in figure 1 and 2. 5, The discussion should include the contribution and limitation of the study.

Author Response

please see atttached word document

Round 2

Reviewer 2 Report

No comments for the revised manuscript.